# Preparation and Properties of Iron Nanoparticle-Based Macroporous Scaffolds for Biodegradable Implants

**DOI:** 10.3390/ma15144900

**Published:** 2022-07-14

**Authors:** Aleksandr S. Lozhkomoev, Ales S. Buyakov, Sergey O. Kazantsev, Elena I. Senkina, Maksim G. Krinitcyn, Valeria A. Ivanyuk, Aliya F. Sharipova, Marat I. Lerner

**Affiliations:** 1Institute of Strength Physics and Materials Science of the Siberian Branch of the Russian Academy of Sciences (ISPMS SB RAS), 634021 Tomsk, Russia; alesbuyakov@gmail.com (A.S.B.); kzso@ispms.tsc.ru (S.O.K.); elena.senkina.1995@mail.ru (E.I.S.); krinmax@gmail.com (M.G.K.); lerner@ispms.tsc.ru (M.I.L.); 2Research School of Chemistry & Applied Biomedical Sciences, National Research Tomsk Polytechnic University, 634050 Tomsk, Russia; vai10@tpu.ru; 3Department of Materials Science and Engineering, Technion, Haifa 3200003, Israel; aliya.f.sharipova@gmail.com

**Keywords:** Fe nanopowder, scaffold, sintering, degradation, strength

## Abstract

Fe-based scaffolds are of particular interest in the technology of biodegradable implants due to their high mechanical properties and biocompatibility. In the present work, using an electroexplosive Fe nanopowder and NaCl particles 100–200 µm in size as a porogen, scaffolds with a porosity of about 70 ± 0.8% were obtained. The effect of the sintering temperature on the structure, composition, and mechanical characteristics of the scaffolds was considered. The optimum parameters of the sintering process were determined, allowing us to obtain samples characterized by plastic deformation and a yield strength of up to 16.2 MPa. The degradation of the scaffolds sintered at 1000 and 1100 °C in 0.9 wt.% NaCl solution for 28 days resulted in a decrease in their strength by 23% and 17%, respectively.

## 1. Introduction

Musculoskeletal system diseases and injuries are the main causes of disability in the population, so their treatment is a global health priority [1]. Implantable devices are widely used to treat musculoskeletal diseases, injuries, and traumas [2]. Traditional technologies have made it possible to develop and obtain implantable materials with greater reliability and durability, but rejection and unexpected degradation of the implants are still serious problems in surgery [3]. Bioresorbable implants are an attractive alternative to traditional permanent orthopedic implants for bone repair [4,5,6,7,8,9]. The advantage of biodegradable implantable devices known as scaffolds is the slow degradation after performing the function of mechanical support of growing bone tissue [10,11].

Scaffolds based on polymers [12,13], ceramics [14,15], metals [16], and composite materials [17,18,19] are currently under development. Metals and metal composites have high plasticity and mechanical strength, which makes them highly promising materials for load-bearing orthopedic implants [20,21]. Among the metals capable of biodegradation in the body (Fe, Mg, and Zn), Fe has significantly higher values of mechanical properties. Fe is also a component of many enzymes and proteins which are necessary for metabolic processes. In neutral and alkaline media, in the case of iron corrosion, cathode depolarization by oxygen occurs and hydrogen is not released, in contrast to the corrosion processes of magnesium alloys [7,22]. In this case, the formed hydroxide (OH^−^) ions interact with the Fe^2+^ ions released into the solution, resulting in iron (II) hydroxide followed by the conversion to iron (III) hydroxide in the presence of water and air oxygen. Numerous studies on cytotoxicity and biocompatibility testify to the possibility of the use of Fe-based materials as biodegradable implants [23,24,25,26]. 

The main disadvantage of Fe implants is their low corrosion rate in the body [27,28,29]. This problem is solved by the use of various additives in the materials that accelerate the biodegradation of Fe implants. Fe-35Mn alloy (35 wt.% Mn), being a solid solution of Mn in γ-Fe (austenite), has been shown to have a higher corrosion rate compared to pure Fe (0.44 mm/yr versus 0.2 mm/yr) [28]. An increase in the corrosion rate of Fe-based alloys has been achieved by adding fine particles of noble metals [29]. These particles act as cathodes in relation to Fe and promote micro-galvanic corrosion, contributing to the active degradation of the Fe matrix. Uniform introduction of Ag nanoparticles on the surface of Fe causes a 2.5-fold increase in the rate of Fe degradation [30]. 

An increase in the rate of Fe corrosion can also be achieved using particles that enhance the anodic dissolution of Fe. Fe oxide is capable of significantly increasing the anodic dissolution of Fe [31]. This can be used to accelerate the degradation of Fe implants. Moreover, Fe oxide, as well as Fe, is a biocompatible species and is widely used in medicine [32].

It should be noted that sintering is one of the obligatory stages in obtaining scaffolds from powder materials [4,7,14,16]. Varying the sintering modes can not only influence the magnetic, structural, and dosimetric properties [33,34] but also significantly change the mechanical properties of the materials [35].

In the present work, the use of an electroexplosive Fe/Fe_3_O_4_ nanopowder to create highly porous scaffolds was investigated for the first time. The nanopowder’s structural, physical, and mechanical properties depending on the sintering modes were studied, as well as the effect of degradation in 0.9 wt.% NaCl solution on the composition, surface structure, and mechanical properties of the scaffolds.

## 2. Materials and Methods

### 2.1. Preparation of Fe Nanopowder

The technique of Fe nanopowder preparation has been described in detail previously [36]. To prepare the Fe nanopowder, an Fe wire (99.8%) with a diameter of 0.3 mm and a length of 65 mm was exploded by a high-current impulse at a voltage of 28.5 kV and a capacitance of 1.6 µF in argon (2 × 10^5^ Pa). The high-current impulse flowed through the wire when the discharge arrangement was operating. Under the action of the current impulse, the wire’s explosive destruction took place. An aerosol containing nanoparticles was taken out of the explosion chamber and precipitated in the hopper. After a nanopowder batch was prepared, the hopper was disconnected from the setup, and the nanopowder was passivated with air oxygen for 24 h.

### 2.2. Preparation of Fe Scaffolds

NaCl particles 100-200 µm in size obtained using the vibratory sieve shaker SS2000 (Powteq, Beijing Grinder Instrument Co., Ltd., Beijing, China) were used as the porogen. The porogen was mixed with the Fe nanopowder in the proportion 61 wt.% Fe/39 wt.% NaCl (the density of the composite was 4.803 g/cm^3^) using a Shaker mixer SM 2.0 Turbula (Vibrotechnik, St. Petersburg, Russia). The stirring time at a bowl rotation frequency of 40 Hz was 30 min. At this proportion, the volume fraction of Fe was 30%. As a result of mixing rounded particles of NaCl coated with Fe, nanoparticles were formed (Figure 1a).

Before sintering, the powder mixture was prepressed on a hydraulic press, T61210B (AE&T, Guangzhou, China), at a sample load of 1.2 N. Two types of samples were prepared: (1) disks 10 mm in diameter and 1 g in weight; (2) beams 60 mm long, 4 mm wide, and 5 g in weight for the three-point bending test. The powder mixtures were sintered in a vacuum furnace, Nabertherm RHTC 80-230/15 (Nabertherm, Lilienthal, Germany), at temperatures of 800 °C, 900 °C, 1000 °C, and 1100 °C. The sintering modes are shown in Figure 1b. Scaffold porosity *P* (%) was determined taking into account the diameter *d* (mm), height *h* (mm), and mass *m* (g) of the sintered samples according to Formulas (1) and (2):(1)$$P=100-\frac{m\times 100}{m_{t}}$$(2)$$m_{t}=\frac{\rho_{Fe}\times \pi \times d^{2}\times h}{4\times 10}$$
where *m_t_* is the mass of the sintered Fe sample at the measured values of *d* and *h*, Г; *ρ_Fe_* is the Fe bulk density (7.874 g/cm^3^).

### 2.3. Experimental Techniques

The morphology and the particle size distribution of the nanoparticles were characterized by transmission electron microscopy (TEM) using a JEM-2100 microscope (JEOL, Tokyo, Japan).

Gravimetric studies were performed using an analytical balance, Acculab ALC-110d4 (Sartorius Group, Johnson Avenue Bohemia, NY, USA), with the geometric dimensions of the sintered samples being measured using a caliper and micrometer.

The phase composition and lattice parameters of the nanopowders and scaffolds were studied using an XRD-6000 diffractometer (Shimadzu, Tokyo, Japan) using CuKα radiation with a secondary monochromator (λ = 1.542Å) at 45 kV and 35 mA. The scans were performed in a 2θ range of 20–90 with a scan speed of 2°/min and scanning time of 1 s. The effect of the scaffold sintering temperature on the crystallite size was determined by plotting the Williamson–Hall equation. The microstructure of the samples and the fracture surface were studied using the scanning electron microscopes Tescan VEGA 3 and LEO EVO 50 (Zeiss, Oberkochen, Germany) equipped with an integrated system of energy-dispersive X-ray analysis (EDX). The size of microstructural elements, pores, and grains of the sintered samples was determined by the intercept method according to ASTM E112. The structure of the scaffolds was also examined by computed tomography (CT) using the X-ray inspection system YXLON Cheetah EVO (YXLON International GmbH, Hamberg, Germany). The three-point bending strength was determined using a DVT GP universal testing machine (Devotrans, Istanbul, Turkey) according to the ASTM E290 method.

### 2.4. Immersion Test

Scaffold degradation was studied in 0.9 wt.% NaCl solution. For this purpose, preweighed samples 60 mm × 4 mm × 5 mm in size were placed in 100 mL of NaCl solution and incubated at 37 °C for 28 days with periodic shaking. The NaCl solution was replaced once a week upon washing the samples with deionized water. After 28 days, the samples were removed, washed with deionized water, and dried at 120 °C for 2 h. After that, the samples were weighed, a three-point bending test was carried out, and the fracture surface microstructure and phase composition were determined.

## 3. Results and Discussion

### 3.1. Nanoparticle Characterization

The nanopowder used to create the scaffold was represented by spherical nanoparticles (Figure 2a) with an average size of 68 nm (Figure 2b). The particles were covered by a 2–5 nm-thick non-continuous oxide layer (Figure 2c). X-ray diffraction measurements revealed the presence of αFe and Fe_3_O_4_ phases (Figure 2d).

### 3.2. Preparation of Scaffolds and Their Characterization

XRD phase analysis of the samples after sintering with a porogenic agent (NaCl) at temperatures of 800–1100 °C showed that at 800 °C, the αFe and NaCl phases were present in the sample (Figure 3a). Increasing the sintering temperature to 900 °C led to the pyrolytic removal of the NaCl pore-forming particles, and the phase composition of the investigated samples was represented by non-stoichiometric Fe oxide Fe_0.918_O [37] and αFe phases (Figure 3b). The samples sintered at 1000 and 1100 °C showed a similar phase composition (Figure 3c,d).

As a result of sintering, the size of the Fe crystallites as determined by the Williamson–Hall method decreased from 99 nm to 65 nm with increasing sintering temperature. The decrease in the size of the Fe crystallites may be due to an increase in the homologous temperature, approaching the temperature range of the sintering mechanism change from solid phase to liquid phase. At the same time, the low duration of isothermal dwelling and the same duration of cooling of the studied samples were insufficient for recrystallization grain growth. This fact can affect the character of the deformation behavior and strength of the samples.

According to the scanning electron microscopy (SEM) data, the scaffold obtained had a porous surface with macropores of different geometries formed due to pyrolytic removal of NaCl particles (Figure 4). The microstructure of the sample obtained at 800 °C was represented by sintered spherical particles of the source nanopowder (Figure 4a, inset). Increasing the sintering temperature led to the coalescence of the particles and the disappearance of interparticle pores. Scaffold sintering at 900 °C resulted in the formation of a grain structure with grain sizes of 5–10 µm, with 1-5 µm particles distributed along the boundaries (Figure 4b, inset). The formation of the oxide occurred along the boundaries of the Fe grains, which is due to the high concentration of defects and the presence of impurities released at the boundaries in the process of prolonged heat treatment [38,39]. According to the data of SEM-EDX analysis in the mapping mode, the particles at the grain boundaries were enriched with oxygen (Figure 5a), which, together with the results of the XRD phase analysis, allowed them to be identified as FeO. Increasing the sintering temperature to 1000 °C led to the growth of Fe grains to ~20 µm and an increase in the size of FeO particles to 4 µm (Figure 4c inset and Figure 5b). At the scaffold sintering temperature of 1100 °C, there was a further increase in the size of the FeO particles to ~10 µm (Figure 4d inset and Figure 5c).

The oxygen content determined by SEM-EDS analysis in the mapping mode did not change significantly and, depending on the sintering temperature of the scaffold, ranged from 4 to 6 wt.% (Figure 5). Thus, the sintering of Fe nanoparticles forming the scaffold resulted in the gradual fusion of Fe particles followed by the formation of large Fe grains. The Fe oxide layer at the surface of the initial nanoparticles moved to the periphery of the formed Fe grains. An increase in the sintering temperature promoted the growth of Fe grains and the sintering of oxide particles.

The porosity of the samples obtained at temperatures of 900–1100 °C was 70 ± 0.8% (Table 1). The decrease in the mass and geometric dimensions of the samples with an increase in the sintering temperature from 900 to 1100 °C may be related to the entrainment of material in the vacuum sintering process.

The shrinkage of the samples in this temperature range also did not change. The pores in the sample sintered at 800 °C were partially filled with NaCl, the share of which was about 25 wt.% (determined by the calculation method). Taking into account the residual porogen content, the porosity of the samples obtained at 800 °C was about 44%.

The average macropore size of the samples corresponded to the average particle size of the porogen used and decreased from 233 µm to 154 µm with increasing sintering temperature (Figure 6). At the same time, when the sintering temperature increased to 1000 °C and 1100 °C, several distribution modes appeared in the histograms (Figure 6c,d). These changes in the pore structure were due to the action of several diffusion mechanisms in the sintering process, leading to an increase in the average size of the micropores as a result of their mutual coalescence, and a reduction in the size of the macropores, being the internal drain of unlimited capacity for vacancies because of their volume shrinkage [40].

Two- and three-dimensional images reconstructed from the XRD tomography (Figure 7) allowed a qualitative assessment of the ratio of material to pores in the scaffold. In the resulting grayscale measurement plane, greater densities of attenuated pixels show brighter regions representing a dense solid. Thus, after pyrolytic removal of the NaCl particles, a strong framework of sintered nanoparticles was formed, being stable at temperatures up to 1100 °C.

### 3.3. Mechanical Properties of Scaffolds

Figure 8 shows typical strain curves of the studied samples as well as the dependences of the three-point bending flexural modulus and the modulus of elasticity on the sintering temperature. The stress–strain relationships of the samples sintered at 800 °C and 900 °C can be characterized as linear-elastic with subsequent brittle fracture, which is consistent with the data of other authors investigating porous metals and cermets [41,42]. The deformation curves of samples sintered at 1000 °C and 1100 °C have a plastic deformation stage followed by a yield stage. The yield strength of the sample sintered at 1000 °C was 14.8 MPa, with that of the sample sintered at 1100 °C being 16.2 MPa.

Although the samples sintered at 900 °C had the highest ultimate strength, they showed the lowest relative strain. Brittle deformable materials are known to be unsuitable for operation under cyclic and shock loads, and the microcracks formed under these operating conditions lead to a drastic decrease in the residual strength. This limits the use of the scaffolds sintered at 900 °C as implants.

SEM observation of the samples sintered at temperatures of 1000 and 1100 °C showed that the microstructure of the fracture surface was characterized by the presence of intercrystalline microcracks and fractures on relatively thin pore walls (Figure 9). Since the investigated samples were prepared by compacting powders followed by sintering at a relatively low Fe homology temperature, this deformation behavior is due to the changing stages of solid-phase sintering, when the formation of interparticle necks and then a strong porous framework is followed by continued consolidation and recrystallization growth of the grain. One would expect that samples sintered at temperatures above 900 °C should have a higher tensile strength. However, particle coalescence and Fe grain growth lead to a dramatic change in deformation behavior, which negatively affects the maximum strength while also increasing the ultimate strain until the yield strength is reached. It should be noted that the bending yield strength of the samples obtained at 1000 and 1100 °C was slightly lower than that of scaffolds with a porosity of 62.3% and significantly exceeded the yield strength of scaffolds with a porosity of 82.3% and of the pure Fe obtained in [21], also corresponding to the parameters of trabecular bone [43].

### 3.4. Degradation of Scaffolds

Intensive corrosion of the samples accompanied by the release of Fe oxides was observed in the process of the degradation of scaffolds sintered at 1000 and 1100 °C in 0.9 wt.% NaCl solution, with the decrease in the weight of the samples over 28 days not exceeding 2%. This is due to the formation and accumulation of degradation products in the form of lamellar structures in the pore space of the scaffolds, which cannot be removed by washing (Figure 10a).

The XRD data show the presence of Fe oxides FeO and Fe_3_O_4_ in the samples. Quantitative XRD phase analysis showed the content of the oxide phases to be 55–65% (Figure 10b).

A decrease in strength properties was also observed for the scaffolds after degradation in 0.9 wt.% NaCl solution. Three-point bending tests showed that the strength of the sample sintered at 1000 °C decreased by 23%, and that of the sample sintered at 1100 decreased by 17%. At the same time, the changes in Young’s modulus were insignificant and within the error range of the values demonstrated by the test specimens before degradation (Figure 11).

The results obtained show the possibility of using this approach to create Fe-based macroporous scaffolds with high mechanical strength. The use of Fe nanoparticles makes it possible to reduce the sintering temperature by 100–200 °C compared to micron-sized Fe particles [4]. Fe oxide isolated on the boundary of Fe grains during the sintering of scaffolds can contribute to the anodic dissolution of Fe in accordance with the mechanism described previously [31] and reduce the degradation time of the material in the body.

## 4. Conclusions

To obtain porous Fe-based scaffolds, an electroexplosive Fe nanopowder and 200 µm NaCl particles as a porogen agent were used. Pyrolytic removal of the porogen from the pressed particle mixture at a temperature of 900 °C and above was found to obtain samples with 70% porosity and high mechanical strength. At the same time, the increase in the sintering temperature to 1000 and 1100 °C contributed to an increase in the ultimate deformation capacity until the yield strength was reached, which was 14.8 MPa and 16.2 MPa, respectively. The degradation of scaffolds in 0.9 wt.% NaCl solution resulted in the oxidation of Fe with the formation of FeO and Fe_3_O_4_ lamellar structures, which amounted to 55–65 wt.% on the surface after 28 days. The bending strength in this case decreased by 23% and 17% for scaffolds sintered at 1000 and 1100 °C, respectively. Among the main advantages of the obtained scaffolds in comparison with their analogues are the high strength properties at a lower sintering temperature and the high porosity. The presence of FeO particles on the surface of scaffolds can accelerate the degradation due to the galvanic effect. To assess the possibility of using the developed scaffolds as orthopedic implants, further studies in the field of medicine are required.

## Figures and Tables

**Figure 1 materials-15-04900-f001:**
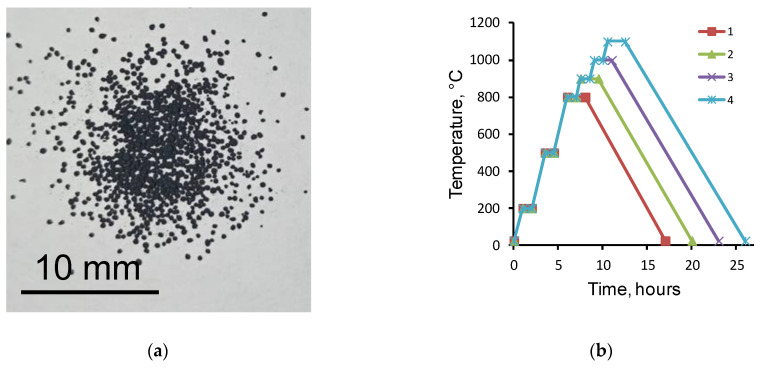
Image of the porogen (NaCl) and Fe nanoparticle mixture (**a**), and sintering modes for the samples (**b**): 1–800 °C; 2–900 °C; 3–1000 °C; 4–1100 °C.

**Figure 2 materials-15-04900-f002:**
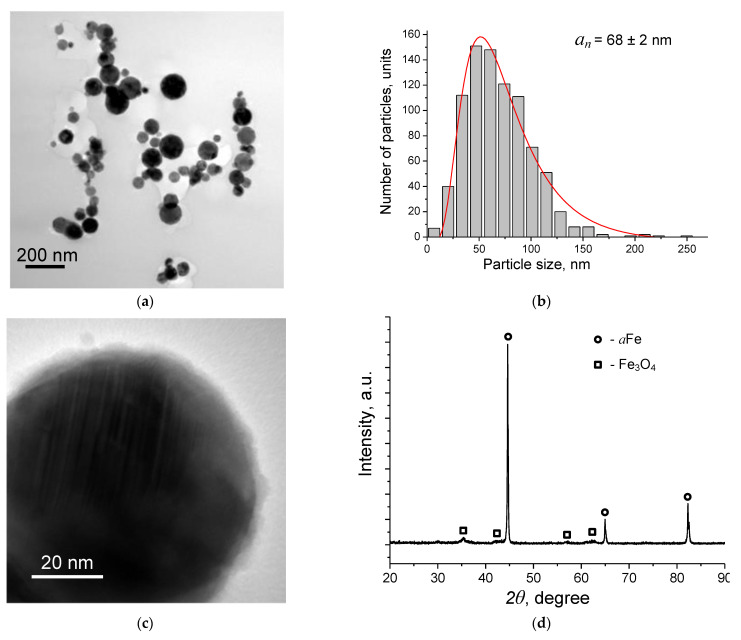
TEM images (**a**,**c**), particle size distribution (**b**), and XRD pattern (**d**) of Fe/Fe_3_O_4_ nanoparticles. The red line on (**b**) corresponds to the log—normal distribution fit.

**Figure 3 materials-15-04900-f003:**
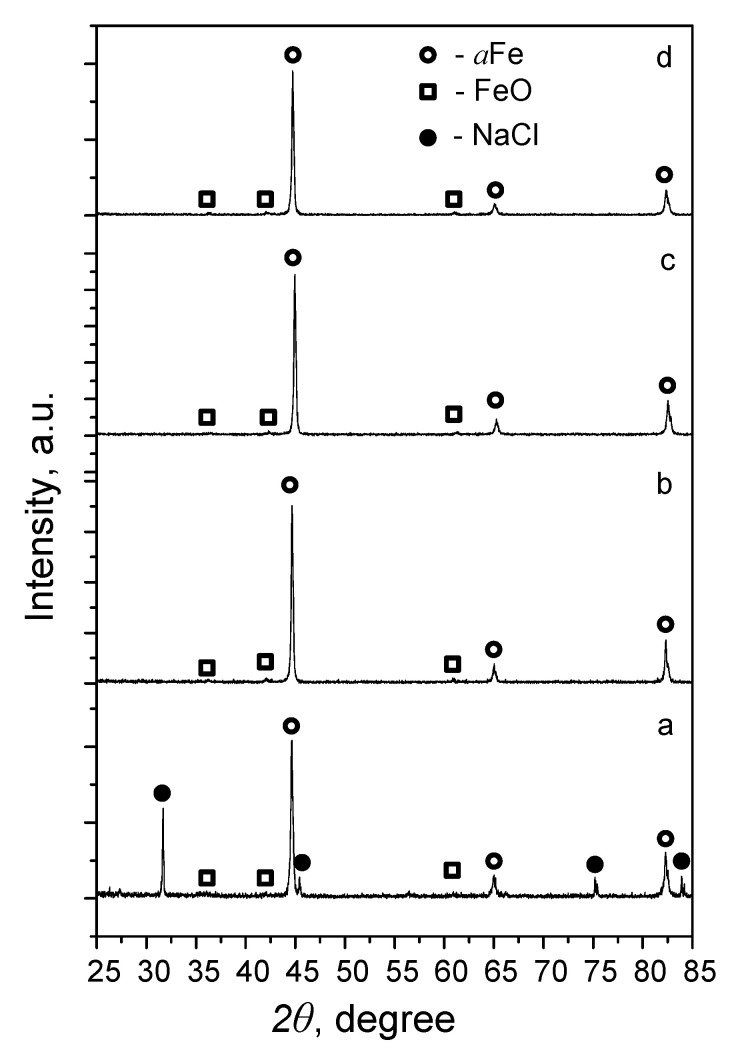
XRD pattern of Fe/FeO scaffolds obtained at a sintering temperature of 800 °C (**a**), 900 °C (**b**), 1000 °C (**c**), and 1100 °C (**d**).

**Figure 4 materials-15-04900-f004:**
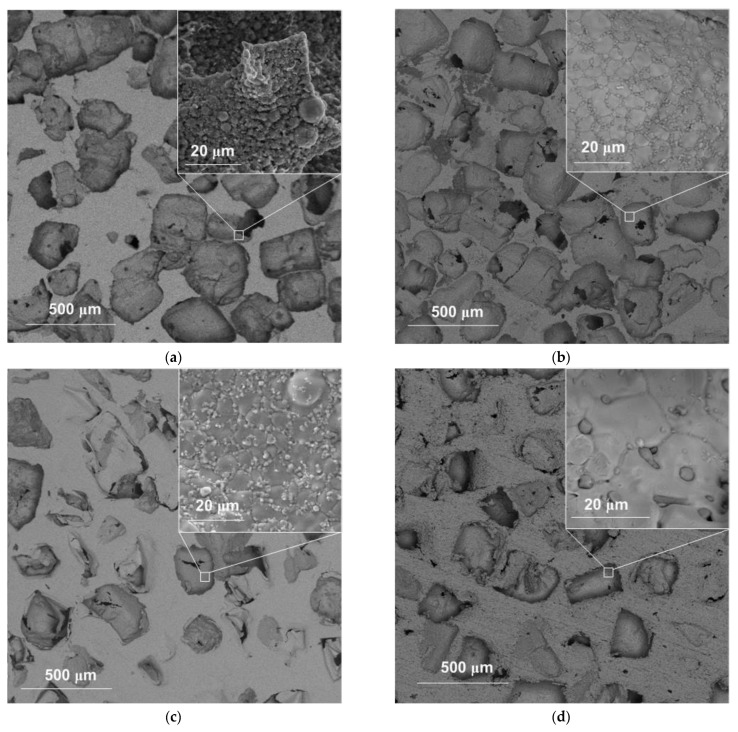
SEM images of Fe/FeO scaffolds obtained at a sintering temperature of 800 °C (**a**), 900 °C (**b**), 1000 °C (**c**), and 1100 °C (**d**).

**Figure 5 materials-15-04900-f005:**
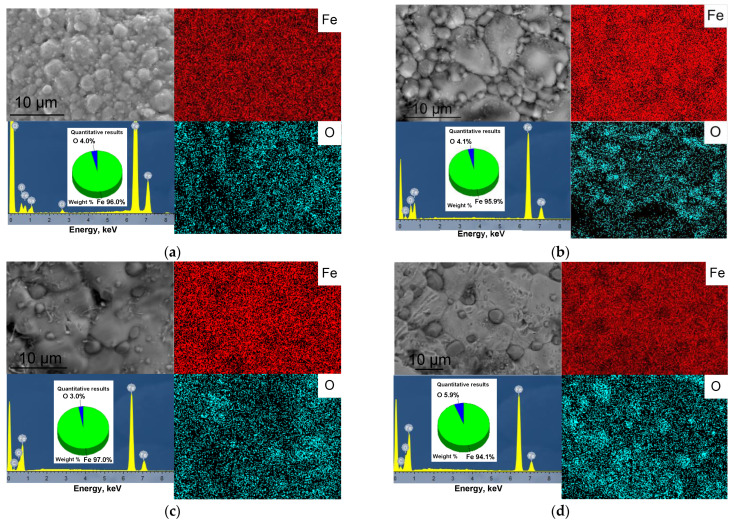
SEM-EDS analysis of Fe/FeO scaffolds obtained at a sintering temperature of 800 °C (**a**), 900 °C (**b**), 1000 °C (**c**), and 1100 °C (**d**).

**Figure 6 materials-15-04900-f006:**
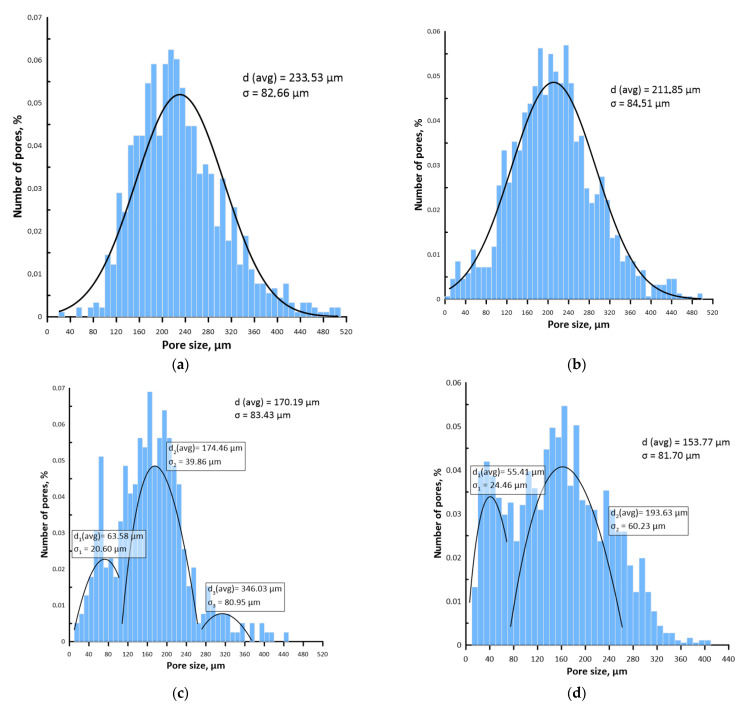
Pore size distribution on the surface of the scaffolds sintered at (**a**) 800 °C; (**b**) 900 °C; (**c**) 1000 °C; (**d**) 1100 °C.

**Figure 7 materials-15-04900-f007:**
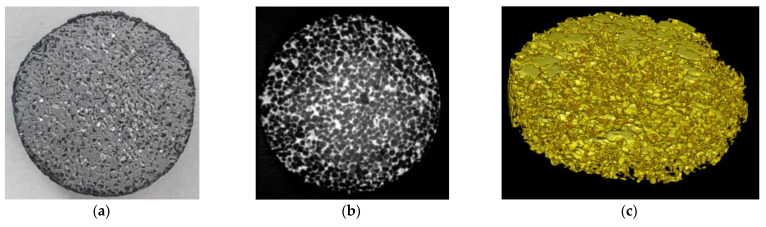
Representation of the image reconstruction via CT. (**a**) Porous sample at 1100 °C, (**b**) 2D cross-sectional images, and (**c**) 3D rendering of the sample. Disc diameter, 9.5 mm.

**Figure 8 materials-15-04900-f008:**
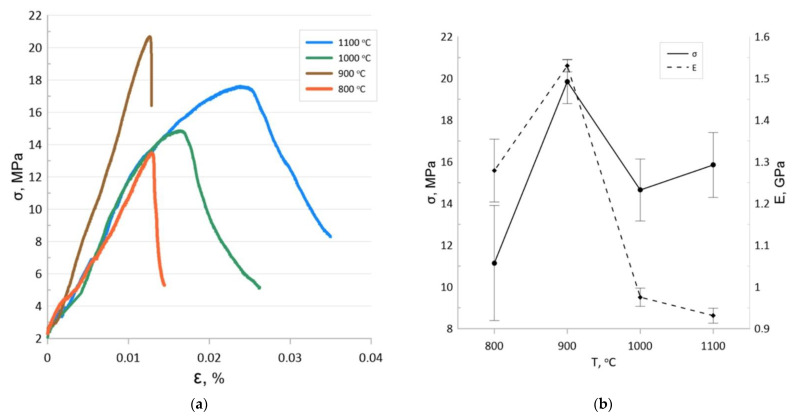
Bending stress–strain curves (**a**) and yield stress and flexural elastic modulus (**b**) of the samples.

**Figure 9 materials-15-04900-f009:**
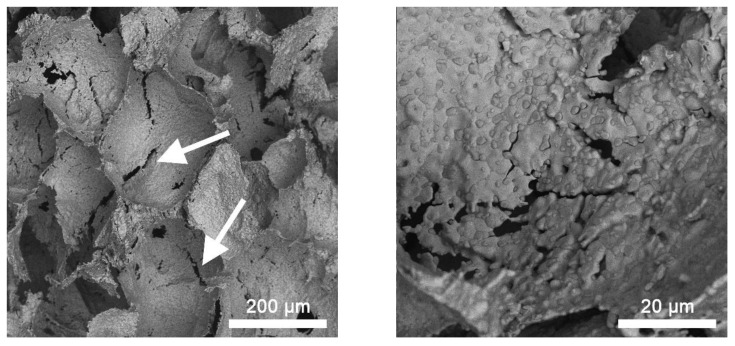
SEM images of the typical fracture surface microstructure of the samples obtained at 1000 and 1100 °C. Arrows indicate intercrystalline microcracks.

**Figure 10 materials-15-04900-f010:**
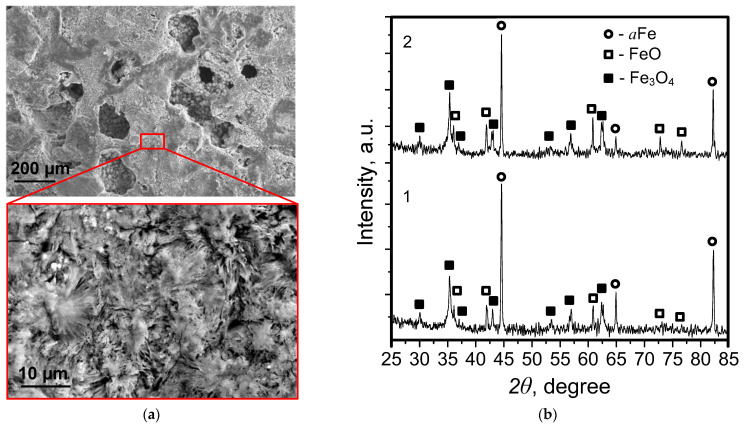
Typical SEM images (**a**) of the scaffold surface after degradation in 0.9 wt.% NaCl solution, and diffractograms (**b**) of the samples sintered at 1000 °C (1) and 1100 °C (2) after degradation in 0.9 wt.% NaCl solution.

**Figure 11 materials-15-04900-f011:**
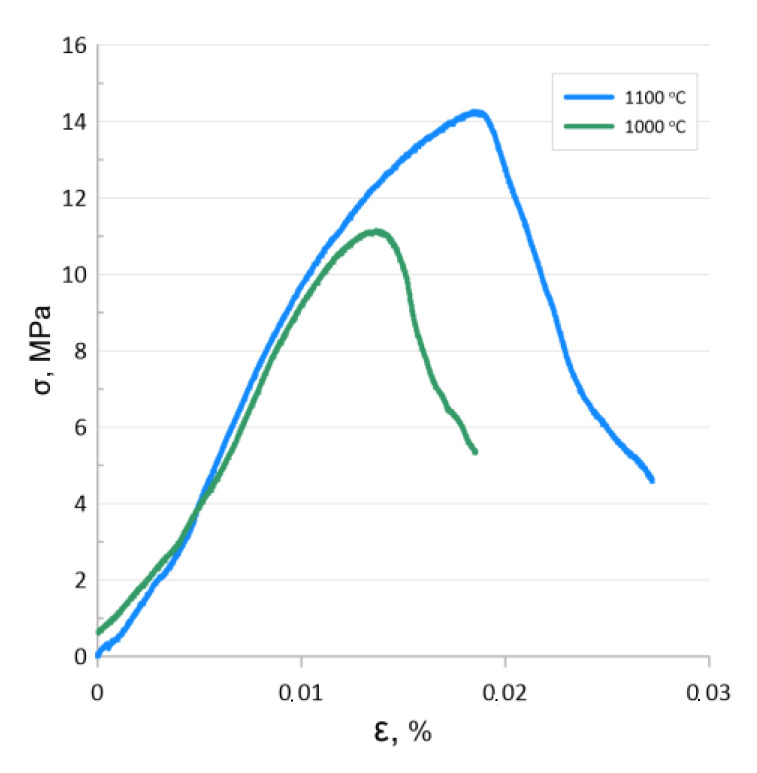
Bending stress–strain curves for scaffolds after degradation in 0.9 wt.% NaCl solution.

**Table 1 materials-15-04900-t001:** Mass, geometric dimensions, and porosity of scaffolds after sintering.

No.	Sintering Temperature	*m*, g	*d*, mm	*h*, mm	*P*, %
1	800	0.815 ± 0.017	9.94 ± 0.03	3.61 ± 0.07	44 *
2	900	0.560 ± 0.014	9.26 ± 0.03	3.43 ± 0.05	69.2
3	1000	0.524 ± 0.012	9.24 ± 0.02	3.40 ± 0.04	70.8
4	1100	0.514 ± 0.011	9.21 ± 0.02	3.27 ± 0.04	70.0

*: porosity including residual NaCl.

## Data Availability

In this study did not report any data.

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
