# Peer review of "Preparation and Properties of Iron Nanoparticle-Based Macroporous Scaffolds for Biodegradable Implants"

_materials, 2022, doi:10.3390/ma15144900_

Round 1

Reviewer 1 Report

The article entitled "Preparation and properties of iron nanoparticle based macroporous scaffolds for biodegradable implants" I have found it to be consistent, clearly presented and in concordance with the scope of Materials MPDI journal.

In the present work the use of electroexplosive Fe/Fe3O4 nanopowder to create highly porous scaffolds was investigated for the first time. Process of obtaining of raw material and characterization of the produced and tested samples are very well and clearly emphasized.

References are adequate and I have appreciated to see that most of these references are dated in 2021-2022, 

The only suggestion I might have for the authors is not focused on the scientific content of the paper, but on the formatting of the paper to be in accordance with the requirements of Materials MDPI journal.

All sections of the article must be counted and I suggest in particular that  the sections entitled "Materials and Methods" and "Results and Discussions" to be divided in sub-sections (even if they will be smaller sub-sections - e.g. subsection related to the producing of nanpowder material, other subsection related to TEM / XRD analysis, other subsection related to SEM-EDS analysis, etc. In this way the article will be much easier to be understanded and followed instead of giving all methods and results in single sections.

Also the format of References provided at the end will have to be re-considered since they are not presented as required / indicated in Materials MDPI Template (instructions for authors).

There are also some minor editing errors in the text (spelling errors) that I have found reading the article - e.g. see the page where figures 7 and 8 are given. The symbol of "degree" Celsius is written without superscript in two places.  These errors are just minor, but I have noticed them.

Ending in a positive way, I have did also one plagiarism checking using the Ithenticate program and I have found one low level of similiarity index, which confirms once again the fact that the article is well written and with high and valuable scientific content.

Taking all these into consideration, my personal opinion is that the article can be accepted / can be published in Materials MDPI journal after doing the minor corrections that I have indicated them to the authors .

Author Response

We appreciate the reviewer for the constructive criticism. We also thank the reviewer for the effort and time put into the review of the manuscript. Major points are highlighted in the text.

Comments and Suggestions for Authors

All sections of the article must be counted and I suggest in particular that  the sections entitled "Materials and Methods" and "Results and Discussions" to be divided in sub-sections (even if they will be smaller sub-sections - e.g. subsection related to the producing of nanpowder material, other subsection related to TEM / XRD analysis, other subsection related to SEM-EDS analysis, etc. In this way the article will be much easier to be understanded and followed instead of giving all methods and results in single sections.

Response: The manuscript has been formatted as required by Materials MDPI journal.

Also the format of References provided at the end will have to be re-considered since they are not presented as required / indicated in Materials MDPI Template (instructions for authors).

Response: The references have been  formatted according to the Materials MDPI Template.

There are also some minor editing errors in the text (spelling errors) that I have found reading the article - e.g. see the page where figures 7 and 8 are given. The symbol of "degree" Celsius is written without superscript in two places.  These errors are just minor, but I have noticed them.

Response:  The errors found have been corrected including symbol of "degree".

Reviewer 2 Report

This paper presents the porous iron-based scaffolds for biodegradable implants. The effect of the particle sintering temperature, composition and mechanical characteristics of the scaffolds was studied. It could provide inspiration for engineers and doctors to design new bioresorbable implants for bone repair. I suggest minor changes to the paper. My detailed comments are as follows:

1.   Please double-check the reference. The format was not correct.

2.   In the Introduction the author says: The process of Fe corrosion in the body occurs without the release of hydrogen. Can you compare with other materials more clearly?

3.   I suggest to use “Fe” or “iron” only. The author often mixed the “iron” with “Fe” in the main text.

4.   In materials and method part, I recommend author to use several sentences to describe the technique of Fe nano-powder preparation.

5.   What is the Powteq? Is that a company name?

6.   Is there any particular reason that the authors choose 61 wt% Fe/39 wt% NaCl while making the porogen? Did the authors compare other proportion? What is the effect of the proportion?

7.   The label and citation of the Figure 1 and 2 is not in the same page. I suggest authors to improve the quality of those figures.

8.   In the figures the units are not written correctly.

9. How many types (size) of Fe particle did the authors used in the work? Due the size effect, the sintering temperature will decrease when the particle size get smaller. Can you add more discussion about the particle size effect?

10. How to control the porous scaffold dimensions?

11. The authors conducted the experiment to describe the mechanical properties of porous scaffold structure. What about numerically analysis?Did authors also use simulations and analyze the phenomena?

12. I recommend authors to describe more about the scaffold dimensions control and the uniformity, if you use “scaffold” in this work. The scaffold often refers the structures with periodic and controllable size.

Author Response

We appreciate the reviewer for the constructive criticism. We also thank the reviewer for the effort and time put into the review of the manuscript. Major points are highlighted in the text.

Comments and Suggestions for Author

  1. Please double-check the reference. The format was not correct.

Response: Literature references have been put in order. The format of the references has been corrected in accordance with the requirements of the Journal.

  1. In the Introduction the author says: “The process of Fe corrosion in the body occurs without the release of hydrogen”. Can you compare with other materials more clearly?

Response: Authors have added a comparison of Fe degradation features with magnesium alloys.

  1. I suggest to use “Fe” or “iron” only. The author often mixed the “iron” with “Fe” in the main text.

Response: the authors have made the necessary corrections, the application of the term has been conformed to the uniformity

  1. In materials and method part, I recommend author to use several sentences to describe the technique of Fe nano-powder preparation.

Response: technique of Fe nanopowder preparation was described in section 2

  1. What is the Powteq? Is that a company name?

Response: Powteq – it's a trademark  of the Grinder Instrument Co., Ltd.

  1. Is there any particular reason that the authors choose 61 wt% Fe/39 wt% NaCl while making the porogen? Did the authors compare other proportion? What is the effect of the proportion?

Response: This ratio has been chosen to obtain scaffolds with a porosity of ~70%, which corresponds to the porosity of the trabecular bone and allows to compare the results obtained with those reported by other authors. 39 wt% NaCl takes 70% of the scaffold volume. The porosity of the samples can be varied in a wide range using this method; accordingly, increasing the amount of porogen we obtain scaffolds with less mechanical strength and vice versa.

  1. The label and citation of the Figure 1 and 2 is not in the same page. I suggest authors to improve the quality of those figures.

Response: This  comment has been corrected

  1. In the figures the units are not written correctly.

Response: This  comment has been corrected

  1. How many types (size) of Fe particle did the authors used in the work? Due the size effect, the sintering temperature will decrease when the particle size get smaller. Can you add more discussion about the particle size effect?

Response: In this work authors use one type of nanoparticles with an average size of 68 nm. The electrical explosion technology makes it possible to obtain nanopowders with a slightly smaller average particle size or to obtain nanopowders with a bimodal particle size distribution. Authors currently conduct research on obtaining scaffolds from nanopowders with a bimodal particle size distribution in order to obtain flowable formulations for 3D printing. Preliminary results show that sintering occurs at the same temperatures of 1000-1100 °C. This will hopefully be the subject of our next publication. Also, there are data in the literature on the formation of scaffolds from micron-sized Fe particles at temperatures of 1200 °C.

  1. How to control the porous scaffold dimensions

Response: To control the size of the scaffolds will be possible by selecting appropriate molds, taking into account the shrinkage of the samples on sintering.

  1. The authors conducted the experiment to describe the mechanical properties of porous scaffold structure. What about numerically analysis?Did authors also use simulations and analyze the phenomena?

Response: Unfortunately, we did not conduct the  numerical analysis, this may be the subject of our further research.

  1. I recommend authors to describe more about the scaffold dimensions control and the uniformity, if you use “scaffold” in this work. The scaffold often refers the structures with periodic and controllable size.

Response:   We have shown that the porosity of scaffolds is determined by the nanopowder to porogen ratio. The sintering temperature of the scaffold samples in the range 900-1100 °C has no effect on the porosity of the scaffolds.  Accordingly, the control of porosity can be carried out by selecting the appropriate ratio before mixing the particles. We also used a fractionated porogen powder in the range of 100-200 μm to obtain a narrower pore-size distribution in the scaffold. The degree of homogeneity of the scaffold can be estimated from the image of the sample obtained by computed tomography (Fig. 7). Authors think that it is not possible to obtain a more ordered structure by this method. It would also like to note that scaffolds with a similar irregular structure were obtained from PLGA, hydroxyapatite, magnesium alloys as described in the papers [Wang, D. X., He, Y., Bi, L., Qu, Z. H., Zou, J. W., Pan, Z., ... & Ding, J. D. (2013). Enhancing the bioactivity of Poly (lactic-co-glycolic acid) scaffold with a nano-hydroxyapatite coating for the treatment of segmental bone defect in a rabbit model. International Journal of Nanomedicine8, 1855. Todo, M., Yos, P., Arahira, T., & Myoui, A. (2018). Development and characterization of porous hydroxyapatite scaffolds reinforced with polymeric secondary phase for bone tissue engineering. in vivo11, 12. Senthilkumar, A., & Gupta, M. (2022). Current and Emerging Bioresorbable Metallic Scaffolds: An Insight into Their Development, Processing and Characterisation. Journal of the Indian Institute of Science, 1-14.]

Reviewer 3 Report

The manuscript “Preparation and properties of iron nanoparticle-based macroporous scaffolds for biodegradable implants” presents the use of electroexplosive Fe/Fe3O4 nanopowder to create highly porous scaffolds. The nanopowder structural, physical and mechanical properties depending on the sintering modes were studied, as well as the effect of degradation in 0.9 wt. % NaCl solution on the composition, surface structure and mechanical properties of the scaffolds. In general, the manuscript is well written and suitable for publication in Materials. Some minor corrections could be taken into account and it needs clarification on certain issues as outlined below.

1)      Have a relook on some grammatical errors available in the manuscript.

2)      Improve Figures quality for the better eminence of your research work.

3)      In section 1;

It should be explained better why sintering work is especially important for this material, and studies on “the effect of sintering temperature” can be mentioned in recent years.

For example;

https://doi.org/10.1007/s40195-018-0795-4

https://doi.org/10.1016/j.nimb.2018.12.036

https://doi.org/10.1016/j.jmmm.2019.165615

4)      In section 2;

a.      There is no information about why they start from a sintering temperature of 800 °C. Have critical temperatures been determined by a thermal analysis such as DSC and TGA?

b.      Density of the materials, intermediate processes, shrinkage in dimensions after sintering should be mentioned.

c.       What is the wavelength of the X-ray used in the XRD device?

5)      In section 3;

There should be a few sentences about the effect of sintering temperature on crystallite size, why wouldn't you calculate crystallinity in such a study?

6)      I appreciate the great effort of the authors; they have prepared this publication beautifully with a lot of analysis and data. However, instead of talking about the data outputs in the Conclusion part, I prefer the advantages and suggestions that the appropriate parameters for the developed material will provide in use.

Taking all these points into consideration, my recommendation is that this article may be published in Materials after a minor revision.

Author Response

We appreciate the reviewer very much for the constructive comments and thoughtful and thorough review. We also thank the reviewer for the effort and time put into the review of the manuscript.The main corrections are highlighted in green in the manuscript. Each comment has been carefully considered point by point and responded. 

Comments and Suggestions for Authors

1)      Have a relook on some grammatical errors available in the manuscript.

Response:  Grammatical errors have been corrected

2)      Improve Figures quality for the better eminence of your research work.

Response:  The figures have been redrawn and authors hope they are improved.

3)      In section 1;

It should be explained better why sintering work is especially important for this material, and studies on “the effect of sintering temperature” can be mentioned in recent years.

For example;

https://doi.org/10.1007/s40195-018-0795-4

https://doi.org/10.1016/j.nimb.2018.12.036

https://doi.org/10.1016/j.jmmm.2019.165615

Response:  The above mentioned studies have been added to Section 1 when discussing the need to study sintering regimes when forming scaffolds.

4)      In section 2;

  1. There is no information about why they start from a sintering temperature of 800 °C. Have critical temperatures been determined by a thermal analysis such as DSC and TGA?
  2. Density of the materials, intermediate processes, shrinkage in dimensions after sintering should be mentioned.
  3. What is the wavelength of the X-ray used in the XRD device?

Response: a.  We did not perform a DSC-TGA analysis for this material. But from our previous experience one can conclude that, despite the absence of noticeable effects at temperatures below the melting point of the materials under study on the DSC-TGA curves, sintering of nanopowders occurs [Lozhkomoev, A. S., Pervikov, A. V., Chumaevsky, A. V., Dvilis, E. S., Paygin, V. D., Khasanov, O. L., & Lerner, M. I. (2019). Fabrication of Fe-Cu composites from electroexplosive bimetallic nanoparticles by spark plasma sintering. Vacuum170, 108980.].  This is confirmed by grain growth, changes in the sample surface microstructure and the acquisition of mechanical strength.

  1. Authors have added the necessary information in Sections 2 and 3.
  2. XRD analysis was performed on an XRD-6000 diffractometer (Shimadzu, Japan) using a CuKα secondary monochromator, CuKα radiation (λ = 1.542Å) at 45 kV, 35 mA. Scans were performed at 2θ 20-90 with a scan rate of 2°/min and a scan time of 1 s. This information has been added to Section 2.

5)      In section 3;

There should be a few sentences about the effect of sintering temperature on crystallite size, why wouldn't you calculate crystallinity in such a study?

Response: The effect of sintering temperature on crystallite size has been described in Section 3.

6)      I appreciate the great effort of the authors; they have prepared this publication beautifully with a lot of analysis and data. However, instead of talking about the data outputs in the Conclusion part, I prefer the advantages and suggestions that the appropriate parameters for the developed material will provide in use.

Response: The Section 4 Conclusion has been revised.